# Care practices, popular knowledge, and health promotion among quilombola woman in Brazil

**Rita de Cássia Moura Diniz[1], Raimunda Magalhães da Silva[1], Christina César Praça Brasil[1], Livia de Andrade Marques[1], Jonas Loiola Gonçalves[2]***

**1** Program Graduate in Collective Health, University of Fortaleza, Fortaleza, Ceara, Brazil, **2** Faculty of Medicine, University of São Paulo, São Paulo, Brazil

* jonasloiola@usp.br

## Abstract

Objective was to investigate the health practices and knowledge among quilombola woman in the 'Baixada Maranhense' region, to understand how these practices are transmitted, transformed, and integrated into well-being and community identity. |Qualitative study was conducted using a hermeneutic–dialectical approach in the Quilombola community of Estiva dos Mafras, Mirinzal, Maranhão, Brazil. Thirteen quilombola woman selected by snowball sampling until theoretical saturation. Data were produced semi-structured interviews period September–December 2024, audio-recorded and transcribed verbatim. The textual corpus was processed in software and thematically interpreted from a hermeneutic–dialectical perspective. Ethical approval was obtained in accordance with Brazilian regulations for social and human sciences research. Five major themes emerged. (1) Health promotion: walking, Zumba, and healthy eating anchored in community life and locally grown foods, alongside spiritual practices for emotional and psychological balance. (2) Intergenerational practices: dialogue as a vehicle for transmitting ancestral knowledge, alongside concern about its erosion, especially the decline of midwifery. (3) Integrative and complementary practices: extensive use of medicinal plants, teas, and "home remedies" combined with biomedical treatments, amid limited recognition by health professionals. (4) Agriculture and healthcare: family farming and extractivism as foundations for healthy eating, income, autonomy, and the inseparability of land, culture, and health. (5) Social movements and territorial legitimation: woman's leadership in community associations as central to securing rights, infrastructure, and territorial recognition. Care practices among quilombola woman articulate body, spirituality, territory, and ancestry, constituting powerful community health technologies that coexist, often tensely, with biomedical care. Strengthening intercultural, intersectional, and territorially grounded health policies requires recognizing quilombola woman as key political and epistemic actors and integrating their knowledge into primary health care and broader health promotion strategies.

**Data availability statement:** Os dados que fundamentam os resultados apresentados no estudo estão disponíveis em file suplmentary.

**Funding:** The author(s) received no specific funding for this work.

**Competing interests:** The authors have declared that no competing interests exist.

## Introduction

The history of quilombola communities in Brazil is marked by resistance and the struggle for survival and autonomy. Originating as refugees for enslaved people who fled farms and sugar mills during the colonial period, quilombos were spaces of freedom and cultural, social, and political resistance [1]. Quilombola communities in Brazil represent a vital legacy of cultural, historical, and social resistance, dating back to the period of slavery. In the 'Baixada Maranhense' region, these communities keep alive traditions that reflect a unique fusion of African, Indigenous, and local influences [1–3].

Among these traditions, health practices occupy a central place, incorporating a deep knowledge of natural resources and a belief system rooted in the relationship between human beings and the environment. Studying these practices is not only a matter of ethnographic record, but also an opportunity to understand how traditional knowledge can contribute to contemporary conceptions of health and well-being [1,3,4].

Contemporary quilombola communities keep this heritage alive, preserving their cultural, social, and economic traditions, while fighting for the recognition of their rights, especially to land. The historical study of these communities reveals not only their origins, but also how health practices have been shaped over time, integrating African, Indigenous, and European knowledge [2,5–7].

Health practices in traditional communities, such as those of quilombolas, are deeply rooted in a holistic understanding of the human being and their relationship with the environment. These practices encompass the use of medicinal plants, rituals, prayers, and other knowledge passed down from generation to generation [7–10]. The effectiveness of these practices is often validated by everyday experience, although in some cases they are also being recognized by conventional medical science. A review of the literature on health practices in traditional communities allows us not only to understand the specificities of these practices, but also to situate them in a broader context of pluralistic health systems [1,2,7,10].

The relationship between traditional knowledge and conventional medicine is complex and multifaceted. While conventional medicine is often based on bio-medical paradigm and scientific evidence, traditional knowledge encompasses a broader understanding of health, which includes physical, spiritual, environmental, and social aspects. The integration of this knowledge into health systems can offer pathways to more holistic and culturally sensitive care practices. The academic literature on this topic discusses both the challenges and the potential of this integration, including issues of validation, preservation of traditional knowledge, and public health policies [2,7].

The territory plays a crucial role in the health practices of quilombola communities, not only as a source of natural resources for traditional remedies, but also as a space of belonging and identity. The relationship with the territory transcends the physical aspect, incorporating symbolic and spiritual dimensions that directly influence health practices. Literature on the geography of health and cultural ecology offers important

insights into how territory, biodiversity, and traditional ecological knowledge interact in the health practices of these communities, highlighting the need for health and environmental policies that recognize and value this relationship [2,7,11].

A time when the global health system faces unprecedented challenges, exploring alternative and complementary health systems becomes crucial. Furthermore, recognizing and preserving this knowledge is urgent given the accelerating loss of biodiversity and cultural changes induced by globalization. Therefore, this research becomes relevant not only to contribute to medical anthropology and public health, but also to health policies that recognize and integrate cultural and knowledge diversity [2].

The assumption is based on the importance of recording, understanding, and appreciating the knowledge and health practices among woman from quilombola communities, which are vital in preserving and transmitting knowledge essential to community health, because their practices induce healthy living in the absence of resources. In this sense, the objective was to investigate the health practices and knowledge among quilombola woman in the 'Baixada Maranhense' region, to understand how these practices are transmitted, transformed, and integrated into well-being and community identity.

## Materials and methods

Qualitative research using a hermeneutic-dialectical approach, prioritizing the understanding of the experiences, perceptions, and practices of quilombola woman in the 'Baixada Maranhense' region in relation to their care practices. Transmissions between generations, understanding how transformative and integrated they are to well-being and community identity. Qualitative research allows for a detailed investigation of phenomena in their natural contexts, emphasizing the meanings that people attribute to them in the face of their social and historical contexts [12].

The hermeneutic-dialectical approach supports understanding, thinking and reflecting on phenomena, seeking a broad and complex understanding of human relations, subjectivities, particularities and socio-historical processes. In this theoretical current, the dialectical movement between thesis, antithesis and synthesis and their structural political and ideological conditioning contradictions emerges [12–14].

Study was conducted in a quilombola community located in the 'Baixada Maranhense' region, named Estiva dos Mafras. This community was selected due to its rich cultural heritage and because it represents a diverse and significant microcosm of the region under study. The community is in the municipality of 'Mirinzal – Maranhão' – Brazil, which is in the northern region of the state, with a territorial area of 686.942 km², a population of 13,978 inhabitants, and a Municipal Human Development Index of 0.62 [15].

Participants comprised 13 quilombola woman over 18 years of age who were interested in participating in the research. The selection of study participants was based on ensuring the representativeness of voices within the community, with a particular focus on those who actively contribute to the dissemination of knowledge and health practices.

Inclusion criteria focused on quilombola woman, over 18 years of age and residing in the community. Women with cognitive or communicative limitations that would prevent understanding the questions or adequate participation in the interviews were excluded from the study. Participants were recruited using the snowball sampling technique. The operationalization of this technique consists of the contribution of a key informant providing indications of possible participants, who in turn indicate new members for the data production process. The participant recruitment phase was finalized by the theoretical saturation of the data, a technique that, from the production of the data, allows visualization for its completion, given that new participants will not bring new theoretical contributions [12].

Data production took place between September and December 2024, mediated semi-structured interviews. The material for data production was based on a flexible semi-structured interview, allowing the exploration of emerging themes according to the reality of the woman, but also containing guiding points for the research objectives. It is noted that a pilot

test was developed with two participants to assess the clarity of the questions and the average duration of the interview, to identify possible adjustments to the interview [12].

Data organization was developed based on semi-structured interviews that were initially transcribed in full, with subsequent grouping in the LibreOffice® Version 6.2.4.2 (x64) program. Subsequently, the consolidated material was processed by the Interface de R pour L Analyses Multidimensionnelles de Textes L de Questionnaires (IRAMUTEQ®) software, version 0.7 alpha 2. The software enabled Descending Hierarchical Classification (DHC), similarity analysis and word cloud construction, with subsequent categorization into classes and identification of words with statistically significant association ($p < 0.0001$). In this sense, the software enabled the identification of statistically significant words, allowing the recognition of occurrence distributions not attributable to chance, which reflected consistent lexical patterns within the analyzed corpus. The p-value expresses the probability that the association between a given word and a specific class occurred by chance, such that values below the adopted threshold indicate greater statistical robustness of this association.

Data analysis was governed by the thematic approach from a hermeneutic-dialectical perspective, seeking to identify, analyze and report patterns (themes) in the collected data. This process will be interactive and reflective, involving initial coding, theme generation, and the review and definition of these themes. It allowed for an understanding of the contradictions and interrelationships between the various themes identified, focusing on the dynamics of change and permanence with regard to the knowledge and health practices of quilombola woman [12].

Regarding ethical and legal aspects, initially a visit was made to the quilombola community of Estiva dos Mafras, located in the municipality of Mirinzal-Maranhão, where the research project, objectives, and procedures of the study were presented, with the presentation of the letter of consent to the president responsible for the quilombola community for her reading and, if she agreed, signature. Subsequently, the work was registered via the Brazil platform, with referral to the Research Ethics Committee of the University of Fortaleza.

The participants included in the study agreed to participate by signing the Informed Consent Form in two printed copies, one for the participant and the other for the principal researcher. Regarding the risks of this study, it presents minimal risks, including embarrassment as a potential risk. From this perspective, to mitigate this risk, the researchers ensure that the study will be conducted in a quiet, calm, and individualized room, creating a welcoming environment that provides comfort to the participants. Regarding the maintenance of the respondents' confidentiality, the codes were listed by quilombola woman, followed by the interview number (e.g., Quilombola woman – 1; Quilombola woman – 2; Quilombola woman – 3…).

Research followed the guidelines set forth in Resolution 510/2016 of the National Health Council, which deals with the ethical precepts of research involving human beings using methodologies from the social and human sciences (BRAZIL, 2016) and was approved by opinion number 6.917–527/2024.

## Results

### Sociodemographic characterization

Population characteristics are a snapshot from the perspective of 13 woman, all self-identifying as quilombola (descendants of runaway slaves) living in a rural area. Regarding age range, there was one young adult (30 years old), four adult woman (44–59 years old), and six elderly woman (over 60 years old). Most of the woman are elderly, highlighting the participation of a 93-year-old woman, which reinforces the importance of intergenerational listening and the valuing of traditional knowledge (Table 1).

Regarding education, most participants were woman with basic education or literate, followed by three with secondary education, two with higher education, and one illiterate woman. Occupations are centered around 11-woman farmers, and only two identify as teachers. The community is predominantly occupied by elderly woman, farmers, and with low levels of education (Table 1).

**Table 1. Sociodemographic characterization of quilombola women, Brazil, 2024.**

| Id. | Age | Education Level | Occupation |
|---|---|---|---|
| 1 | 93 | Illiterate | Farmer |
| 2 | 61 | Basic education | Farmer |
| 3 | 66 | Elementary education | Farmer |
| 4 | 30 | Elementary education | Farmer |
| 5 | 67 | Elementary education | Farmer |
| 6 | 57 | High school | Farmer |
| 7 | 54 | High school | Farmer |
| 8 | 69 | Elementary education | Farmer |
| 9 | 73 | Elementary education | Farmer |
| 10 | 44 | Higher education | Teacher |
| 11 | 86 | Literate | Farmer |
| 12 | 44 | Higher education | Teacher |
| 13 | 60 | High school | Farmer |

## Analysis of the textual corpus

The analysis of the corpus, composed of 13 texts, revealed a total of 655 segments, in which 2,988 forms and 22,274 occurrences were identified. The average recorded was approximately 34 forms per segment, evidencing a moderate lexical density. Among these forms, 2,016 lemmas were recognized, corresponding to the lexical units in their canonical form. Segmentation and classification resulted in five themes from the six classes, which are grouped with their respective sub-themes, given the utilization rate of 82.90% of the processed material.

### Theme 1: Health Promotion

**Sub-theme 1: Physical exercise and healthy eating.** Health promotion from the perspective of quilombola woman is experienced through the practice of physical exercise in the community environment, with walking being developed and, in conjunction with this, the pursuit of a healthy diet as an act that promotes health. In this way, some also point to the practice of Zumba as a measure to promote physical well-being in the community context.

> We go for walks, we used to have Zumba, but we stopped more because of the alcoholism issue. But we used to have a lot of dance parties, but we stopped... so we take advantage of everything in this space. Walking is much healthier for us; with so many plants, we've learned not to burn them (Quilombola woman 2).

> We have our walks, our work, all that busyness of doing things. We shouldn't be idle because it's bad to be idle (Quilombola woman 7).

> I walk in the morning, three times a week, to help me maintain my health and try to eat healthy things; we eat mostly fish (Quilombola woman 3).

> We do physical exercise, walking. Besides that, there's no other way for us to exercise. Promoting health is more about food, since it's very healthy (Quilombola woman 12).

**Subtheme 2: Promoting Health Through Spirituality.** Health promotion is highlighted using practices that involve spiritual dimensions, revealing the use of spiritual measures as an act that promotes health for self-knowledge and self-help, but also for other people who seek the spiritual resources promoted by the community. This practice is experienced as an alternative measure to promote health, even though there is prejudice and misinformation about spiritual practice in

the perception of these communities. However, this perception reaffirms the importance of promoting spiritual health for oneself and for others within and outside the community.

> My practice is that I am from the Umbanda religion, and in our religion we often say that whether you come with money or without money, the important thing is to do good, to help others... so we try to help not only the people of the community but also people from outside, even though it is a religion that is very criticized, with a lot of prejudice even today, but we always try to help and give our best. We offer spiritual treatments for psychological healing; sometimes a person arrives with an obsessive spirit that keeps disturbing them... I help some people who arrive, I can say, who are spiritually ill, I try to help (Quilombola Woman 12)

> ...you arrive there in Estiva and we take very good care of the environmental issue. My grandmother, my grandmother was from African-based religions, you know what that is... she had a gift and she had knowledge of herbs, she prayed, she guided people, so she was from African-based religions, she had knowledge, but her knowledge was that she cleansed people (Quilombola Woman 1)

> In our spiritual work, we use a lot of elements of nature, herbal baths for cleansing if you are in pain. For a headache, a bath with leaves from a fragrant plant to relieve headaches, folk healing practices, this type of practice (Quilombola Woman 12)

## Theme 2: Intergenerational Practices

**Sub-theme: Intergenerational transmission of ancestral knowledge.** Quilombola woman recognize the importance of dialogue as a promoter of the transmission of ancestral knowledge from generation to generation, demonstrating that this practice of belonging within the community is important knowledge for the promotion of cultural identity. In this sense, they seek to establish this knowledge as the know-how and practice belonging to their history, a recognized link for the affirmation of their roots, especially their ancestral African roots.

> This knowledge, transmitted from generation to generation, in our case through conversation and dialogue about the practice itself, we talk and then show what really happens in practice so that they can learn... The practice we have here in the community is very important because it is knowledge that is passed down from generation to generation, and we young people accept this agreement of knowledge (Quilombola woman 2).

> It's about us effectively passing on knowledge from generation to generation, knowing and understanding your history, your roots, and using this knowledge to pass it on to them as a guarantee (Quilombola woman 1).

**Sub-theme 2: Loss of the transmission of ancestral knowledge.** Quilombola woman express their concerns regarding attempts to transmit ancestral knowledge to new generations, seeking to pass it on to future generations, but at the same time highlighting a loss in the transmission of this knowledge to other generations, recognizing that future generations will experience this loss of knowledge.

> The knowledge passed down from generation to generation; we try to transmit it to future generations. We can even transmit it if, for example, I have a child, and if we talk about it, they think it's nonsense (Quilombola woman 6).

> It had something else, oh my god, I forgot, now it was arueira, that thing that closed off these practices, they are not being passed on to the new generations, they are not being passed on as they should be (Quilombola woman 7)

**Sub-theme 3: Loss of the midwives' legacy.** Faced with this loss of intergenerational knowledge transmission, quilombola woman are experiencing a breakdown in this transmission, especially among midwives, once an essential resource for childbirth and delivery. The transmission of this knowledge is broken, and an essential practice in woman's care is lost. The participants also acknowledge that, within the biomedical model, midwives provided security during normal childbirth in the community.

> In the hospital, given the complexity, there are midwives here. When things don't work out here, they go to the main hospital where there's a midwife. There's an experienced midwife there... When there's no passage, she'll do a cesarean section. Until then, the baby stays with the midwife until it comes out naturally (Quilombola woman 3)

> It gave a sense of security: you do this, you do that, you do that, and then you give birth. I gave birth with midwives like that. My mother has delivered many babies, my sister knows how to deliver babies (Quilombola woman 1)

> There are no more midwives now, after she died my aunt, one of her daughters was also a midwife... all the midwives died, there are none left... (Quilombola woman 5)

## Theme 3: Integrative and complementary practices in health promotion

**Sub-theme 1: Medicinal herbs in promoting care.** Integrative and complementary practices are allies in promoting the care of Quilombola woman, in which the use of medicinal teas is an ancestral practice that impacts the physical and mental well-being of the participants. Those who have overcome the challenge of using tea as an alternative for comprehensive health care, even in urgent situations, seek through integrative practices – tea – its use as a traditional and essential resource within their lived context.

In this sense, resources are combined with traditional medicine and cultural, social, and environmental aspects. It is noteworthy that the pursuit of traditional medicine is allied with alternative medicine, with the use of tea being unified with 'pharmacy medicine' in the search for healing and the promotion of well-being within the community context.

> As they pass by, one teaches you tea in line, you 're in line today for medicine, and they're handing out medicine. There are people from my time, some new ones, and one who taught me. There's the gapou strip, and they've already finished the sticks that need to be given for everything, it's all there, each one is teaching a tea, and I, at least since I had congestion, haven't taken injections... (Woman quilombola 13)

> We have an ambulance, people, we have an ambulance to take you, but if you're really in pain inside, then we make a tea to relieve it... if it goes away with the tea, we drink the tea and stay there, if it gets worse we have to find a way outside, sometimes they come to do it in Mirinzal (Woman quilombola 11)

> Lemongrass tea is great as a calming agent and also very good for the stomach. There's also gardenia, which is also very good for the heart, and there are numerous others (Woman quilombola 2)

> We take pharmacy medicines, as we call them, but we also have a tea for diabetes, we have a tea to lower blood pressure for those with hypertension, we have a tea for itchy skin – we call those itchy skin rashes – to take a bath with, so we have all these traditional remedies that we cultivate in the community, but we also respect those that the agent of health (Woman quilombola 2)

> When the pain is very old, we make tea from Santa Maria, which is a type of wood that has to be bitter when a woman has a baby that is born with a tooth; that's how it was in my time (Woman quilombola 11)

Tea practices include various teas; there's a tea we make from neem leaves that's good for stomach migraines. You can add peppermint, which is a small mint plant that we use to make tea and drink it; it provides relief (Quilombola woman 11)

**Subtopic 2: "Home remedies" as a care alternative.** The cultural and historical practice of promoting community care is experienced through integrative medicine, which promotes the use of 'home remedies' in the face of illness. These care practices are perceived and combined with the use of prayer in the pursuit of holistic care.

Any issue like flu, cough, body aches, even if someone fractures a bone, there are remedies available, and regarding folk healers, we still have a young woman (Quilombola woman 8).

We take our own precautions, so we have several [remedies]. We still take many home remedies, which were the teachings of the old woman that she left for us, and we continue to use them to this day (Quilombola woman 1).

Amidst alternative practices and popular knowledge, quilombola woman experience a lack of integration between traditional and alternative medicine, which they perceive as a lack of cultural and traditional knowledge on the part of health professionals, stemming from a lack of guidance on the use and importance of 'bush remedies'.

"They don't give out herbal remedies, they tell you to buy this box of injections or buy this pill, they respect that, but they don't give them out, they don't know, it's because they don't know" (Quilombola woman 13).

He comes from house to house [health worker] to find out if we want a consultation, if we need a consultation, and sometimes the medicines that we have, sometimes we have home remedies (Quilombola woman 7).

**Theme 4: Agriculture and healthcare**

**Sub-theme 1: Family farming in promoting healthy eating.** Quilombola woman carry with them the culture of family farming, a dimension that intersects with healthcare through the promotion of products for care and the promotion of healthy eating. Family farming intersects with the promotion of natural cultivation practices without compounds that harm human health. The promotion of care is a result of extracting the essential and natural products from the crops to promote acts of care; family farming promotes the production of teas and the subsistence of the community, since the planting is diverse and incorporates social practices of paramount importance for healthy habits.

I have, I give you, ma'am, you have, give me, you who work hard in the communal work, so that when you go to work with the other, you already have it there, while I haven't yet done my field work or made my flour, and you have, you already give me, so you plant...wherever you arrive there's something to eat, there's chicken, there's ora-pro- nóbis, which we've now internalized, ora-pro -nóbis is a plant from Minas Gerais that is good for many different types of illnesses (Quilombola woman 8)

The land for us is the usufruct, what I have on it, what I value in it, it's my possession, it's my home, it's my plant... we work with our organic fertilizer, so we don't even use our cassava cuttings that you plant, but we don't put any dye in our dough to make a yellow flour... it's the seed that we put there, it's the yellow one, it's natural, we don't use anything on our food, on that, the juçara palm, we work with it today, we already plant it (Quilombola woman 1)

We learned to do crop rotation, which is planting one crop after another. We plant beans... coconut, which we also plant, cashew, Jussara, and then we have fishing, which is also artisanal and natural... we plant watermelon. Nowadays we've learned to plant watermelon on a large scale, so we already have a production that we pass down from

generation to generation. We harvest the beans, take the bean husks, leave them there, plant corn, and then plant them again. (Quilombola Woman 2)

We have our vegetable garden to grow our plants, we have our yards to sweep, so everything is... and at first we used to break a lot of coconuts, we spent the day breaking coconuts, today nobody wants to break coconuts anymore, right? My mother used to break coconuts to make oil (Quilombola woman 7)

All of this, for example, saffron, we plant it today because we know that saffron is anti-inflammatory, so we have it naturally and we consume it (Quilombola woman 2)

**Sub-theme 2: Family-based extractivism in community empowerment.** Natural resources in the community are perceived as intrinsically linked to human beings and nature, where female empowerment is permeated by family farming production and income generation. These mechanisms are enhanced by the support of social food acquisition programs; while for some agriculture is a means of subsistence, for others it is a supplement to their income, generating female empowerment and the sharing of natural products with other communities.

In this sense, one of the statements reveals a sense of fulfillment in extractive practices focused on family farming, as a quilombola woman remarks, '*it's wonderful to live in the countryside*'.

We have many natural resources from the community that we extract food from; we have fishing grounds, the people in the community also fish, we raise chickens, we raise pigs, some (Woman quilombola 12)

In our fishing cultures, everything left by them is our survival, which was based on extractive activities, such as extracting cassava, fishing, and harvesting buriti and açaí (Woman quilombola 6)

We have a government program. The federal government is buying our product, which comes from an extractive community. We extract the juçara palm, then it goes to CONAB (National Supply Company), and CONAB donates it to CRAS (Woman's Social Assistance Reference Center) (Woman quilombola 2)

Some raise cattle in their backyard, and so on. There's also 'açaí', which is extracted directly from the backyard, which is very natural, so it has a lot of iron and a lot of protein. It's wonderful to live in the countryside, it's wonderful! (Woman quilombola 12)

## Theme 5: Social movements and the legitimization of quilombola territories

**Sub-theme: Community association in promoting the rights of the Quilombola population.** Social movements for quilombola woman are perceived as a link to empowerment and the construction of territories demarcated by equality. Through the consolidation of a community association, they realize that female community leadership is the essential pillar for guaranteeing rights and promoting equality within the living territories of the quilombola people, enabling the demarcation of territory and the consolidation of community dreams, with an emphasis on basic social rights such as food, sanitation, and housing.

She has always worked very hard for the association, everything she does is related to other social movements and leadership that she has, even political party involvement, so she seeks and brings in income, financial support through the association, and the search for, let's talk about, housing, it's not a housing issue, it's not about rights, it's about social services (Quilombola woman 2)

All the benefits within the community, we have the association, which is our organization. Everything we can see, the association has provided. We had water, then we had septic tanks... first I saw the septic tanks, then every house with a small bathroom inside the house in the yard, electricity, roads, everything was through the association. We now have 50 houses from the "My House" project (Quilombola Woman 6).

## Discussion

The results reveal that health promotion and care practices among quilombola women are experienced through an expanded conception of health, encompassing bodily practices, spirituality, ancestral knowledge, family farming, and community activism. From this perspective, community care is articulated with a critical understanding of the social determinants of health, in which living conditions, cultural practices, territorial dynamics, and power relations are understood as interrelated processes that shape the production of well-being and the health–disease process, particularly among socially marginalized and historically vulnerable populations [16–19].

In this sense, when examining the social determinants of health that intersect with this community, it is essential to consider their socio-historical values, recognizing that these determinants should not be employed in an uncritical or merely descriptive manner, as such an approach may obscure power relations, institutional inequalities, and historical processes that produce and reproduce health inequities. Accordingly, the analysis developed here is grounded in a framework that goes beyond the enumeration of social factors, proposing a relational and contextualized understanding articulated with gender, race, territory, and the lived experiences of quilombola women [13,14,18].

In this sense, the production of care has historically been occupied by woman, and they play central roles in preserving traditions and cultural models essential for the legitimation of Quilombola territory [13,14]. Conversely, national and international evidence reveals numerous inconsistencies, due to structural and social determinants that have caused the disruption of internal generational flows and of knowledge, skills, and practices for maintaining the living territory and the Quilombola population, mainly due to cultural fragility, loss of autonomy, and the vulnerability that still exists due to the absence of effective and consolidated public policies in the Brazilian reality [20–25].

Therefore, the data produced highlight the complexity and the need for public policies to guarantee timely access for this population, given that even with guaranteed constitutional rights, the living conditions of quilombola woman are still fragmented and disrupted in the face of the logic of care. Even though the Brazilian government has specific policies for this population, these are poorly consolidated and established in the face of Brazil's territorial and social dimensions. In line with this problem, the capillarity and fragility of primary health care for adequate, timely, comprehensive, and longitudinal follow-up still hinders health care [20,26–28].

The weakening of care is exacerbated by the absence of care practices that recognize socio-historical contexts and their integration with the health sector, resulting in a non-dialogical practice between the real and the essential aspects of care for this population. This marginalization produces and reinforces socio-historical inequalities and the structural patterns of environmental and institutional racism that this population still experiences in Brazil [24,25,28].

In this context, robust and consolidated policies are essential for the promotion of care in Brazil, especially those that recognize and integrate the production of care as a logic between popular and biomedical knowledge. This complexity is fundamental to understanding within socioeconomic, political, and cultural fields so that the transmission of knowledge and care occurs in an equitable and comprehensive manner [8–10,26,27].

In the neoliberal model, the colonial process of knowledge delegitimizes Black, feminine, and community knowledge, thus producing silencing and invisibility of powerful care practices that are essential for community care and the population's sense of belonging. This leads to the silencing and invisibility of powerful practices of care, healing, and management of bodies and territory [29–31].

Among quilombola woman, this translates into a reduction in traditional practices of midwifery, blessing, use of medicinal plants, community rituals, and ancestral care strategies. In this sense, it is essential to have public practices and policies that recognize the participation of quilombola woman in decision-making processes. The various programs and actions must incorporate an intersectional perspective, recognizing the diverse interfaces that permeate care, especially one that is able to recognize feminine, popular, territorial, social, and systemic knowledge, in which care is built in an integral, in-depth way and with an epistemological cultural continuity [4,16,30,32–34].

In this sense, promoting the continuity of strategies and practices of the living territory are essential mechanisms for maintaining resistance and re-existence. In which the integrative and complementary practices developed from medicinal plants, woman's circles, transmission of popular knowledge about childbirth, spiritual rituals, and community activism are powerful articulations for maintaining identity and care practices. We reinforce the essence of this knowledge from an intersectoral and participatory perspective, focusing on the interculturality and intersectionality of knowledge so that we can build a sustainable world [16,28,32,35].

In this sense, health care in quilombola communities is permeated by the need for a holistic perspective, not restricted to biomedical indicators or the provision of essential services. An intersectional and human-centered approach is essential, fostering connection with ancestry, spirituality, social activism, and the very production of knowledge. Recognizing and incorporating these dimensions into public health policies and professional practices is essential for building a more sustainable world aligned with the unique forms of resistance and care of the female population [29,32–34].

## Conclusion

The promotion of health among quilombola woman is experienced through the regular practice of physical exercise, such as walking and Zumba, combined with a healthy diet based on the consumption of natural foods grown by the community. These activities not only strengthen physical well-being but also reinforce community life and contact with the natural environment, valuing habits that prevent sedentary lifestyles and promote quality of life.

The spiritual dimension constitutes an important axis of care for quilombola woman, who use religious practices of African origin, blessings, and herbal baths as ways to promote physical and psychological health. Despite external prejudice, these practices are recognized internally as essential for balance and emotional support, highlighting spirituality as a therapeutic resource integrated into daily community life.

Intergenerational dialogue appears as fundamental for the transmission of ancestral knowledge, which strengthens Quilombola cultural identity and the relationship with its African roots. However, there is growing concern about the loss of this traditional knowledge, especially regarding midwives, whose practices, once essential for maternity and childcare, are declining in the face of the influence of the biomedical model and social changes.

Integrative and complementary practices, especially the use of medicinal teas and home remedies, reinforce comprehensive care in the community, combining traditional knowledge and modern medicine. These practices reflect ancestral knowledge that remains alive and functional, although there is some distance or lack of awareness on the part of conventional health professionals, limiting the integration of knowledge. Finally, family farming and extractive activities represent fundamental pillars in promoting health, ensuring healthy, sustainable food and sources of income for quilombola woman.

These practices support community and woman's empowerment, strengthening territorial autonomy and the deep connection between care for the land, culture, and collective health. Social movements and community associations play a vital role in defending these rights and consolidating quilombola territories.

## Supporting information

**S1 File. Full quotes and interpretations.**
(DOCX)

## Author contributions

**Conceptualization:** Rita de Cássia Moura Diniz, Raimunda Magalhães da Silva, Christina César Praça Brasil, Jonas Loiola Gonçalves.

**Data curation:** Rita de Cássia Moura Diniz, Raimunda Magalhães da Silva, Jonas Loiola Gonçalves.

**Formal analysis:** Raimunda Magalhães da Silva, Christina César Praça Brasil, Livia de Andrade Marques, Jonas Loiola Gonçalves.

**Investigation:** Rita de Cássia Moura Diniz.

**Methodology:** Rita de Cássia Moura Diniz, Raimunda Magalhães da Silva, Livia de Andrade Marques, Jonas Loiola Gonçalves.

**Resources:** Jonas Loiola Gonçalves.

**Supervision:** Rita de Cássia Moura Diniz.

**Visualization:** Livia de Andrade Marques.

**Writing – original draft:** Raimunda Magalhães da Silva, Christina César Praça Brasil, Livia de Andrade Marques, Jonas Loiola Gonçalves.

**Writing – review & editing:** Raimunda Magalhães da Silva, Livia de Andrade Marques, Jonas Loiola Gonçalves.

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
