## [Decision Letter · Decision Letter 0]

15 Jan 2026

Dear Dr. Gonçalves,

Thank you for submitting your manuscript to PLOS ONE. After careful consideration, we feel that it has merit but does not fully meet PLOS ONE’s publication criteria as it currently stands. Therefore, we invite you to submit a revised version of the manuscript that addresses the points raised during the review process.

We look forward to receiving your revised manuscript.

Kind regards,

Rafael Galvão de Almeida, PhD.

Academic Editor

PLOS One

**Journal Requirements:**

1. When submitting your revision, we need you to address these additional requirements. Please ensure that your manuscript meets PLOS ONE's style requirements, including those for file naming. The PLOS ONE style templates can be found at https://journals.plos.org/plosone/s/file?id=wjVg/PLOSOne_formatting_sample_main_body.pdf and https://journals.plos.org/plosone/s/file?id=ba62/PLOSOne_formatting_sample_title_authors_affiliations.pdf 2. Please include your tables as part of your main manuscript and remove the individual files. Please note that supplementary tables (should remain/ be uploaded) as separate "supporting information" files. 3. Please include captions for your Supporting Information files at the end of your manuscript, and update any in-text citations to match accordingly. Please see our Supporting Information guidelines for more information: http://journals.plos.org/plosone/s/supporting-information. 4. If the reviewer comments include a recommendation to cite specific previously published works, please review and evaluate these publications to determine whether they are relevant and should be cited. There is no requirement to cite these works unless the editor has indicated otherwise. 

Reviewers' comments:

**Comments to the Author**

1. Is the manuscript technically sound, and do the data support the conclusions?

Reviewer #1: Yes

Reviewer #2: Yes

2. Has the statistical analysis been performed appropriately and rigorously?

Reviewer #1: No

Reviewer #2: Yes

3. Have the authors made all data underlying the findings in their manuscript fully available?

Reviewer #1: No

Reviewer #2: Yes

4. Is the manuscript presented in an intelligible fashion and written in standard English?

Reviewer #1: Yes

Reviewer #2: Yes

**Reviewer #1:**

Comments to the Authors

I appreciate the opportunity to review this manuscript, which addresses a relevant and socially sensitive topic by discussing care practices, popular knowledge, and health promotion among quilombola women. The study presents important merits, such as the valorization of traditional knowledge, the timeliness of the references, and the adoption of a qualitative approach consistent with the object of investigation. However, some methodological, conceptual, and editorial aspects need to be revised to strengthen the clarity, internal consistency, and transparency of the manuscript.

General Comments

In general, the manuscript would benefit from greater methodological precision and adjustments in the presentation of results, especially regarding the description of analytical procedures, the consistency between text and presented materials, and the explicit mention of some theoretical frameworks used throughout the text. Major Comments

Redundancy in the description of the hermeneutic-dialectical approach:

In the Materials and Methods section, two consecutive paragraphs describe the hermeneutic-dialectical approach in a very similar way, reiterating the thesis-antithesis-synthesis movement and the socio-historical contradictions. It is recommended to condense these passages into a single, more concise paragraph in order to avoid conceptual repetition and improve the flow of the methodological section.

Insufficient description of the statistical dimension of the textual analysis (IRAMUTEQ):

The authors state that the analysis in IRAMUTEQ allowed the identification of words with statistically significant association (p < 0.0001). However, they do not explain which statistical procedures were used (e.g., chi-square test), nor the meaning of this p-value in the context of Descending Hierarchical Classification, nor how these associations supported the interpretation of the results. It is recommended to detail these aspects to ensure greater transparency and reproducibility of the analysis.

Mention of a table missing from the manuscript:

The text refers to “Table 1” in the sociodemographic characterization of the participants; however, this table was not included in the submitted file nor presented as supplementary material. The inclusion of Table 1 or a revision of the text is requested, ensuring consistency between the body of the manuscript and the materials provided.

Exclusion criteria and potentially stigmatizing language:

The exclusion criterion described as “women with mental health-related needs” can be interpreted broadly and in a stigmatizing way. It is suggested that the passage be reformulated, explicitly stating that the exclusion refers to cognitive or communicational limitations that would prevent the understanding of the questions or adequate participation in the interviews, and not to the mental health condition itself.

Minor comments:

Duplicate references:

Duplication of the same reference was identified, listed twice (refs. 22 and 28). It is recommended to unify the referencing and adjust the numbering of citations throughout the text.

Conceptual precision in the use of some references:

The reference to Minayo (2021), which presents a critique of the uncritical use of the concept of "social determinants of health," is used in the text without explicitly stating this critical position. Greater conceptual precision is suggested when engaging with this work.

Concepts mentioned without direct reference:

Some relevant concepts such as environmental racism, institutional racism, and coloniality of knowledge appear in the text without explicit reference in the corresponding paragraphs. It is recommended to include specific citations that support these statements.

Use of broad theoretical frameworks that are not fully explored:

Classic authors such as Habermas and Gadamer are cited to support the hermeneutic-dialectical approach, but are not analytically revisited throughout the text. It is suggested to evaluate the need to maintain them or deepen their articulation with the empirical analysis. Final Considerations

The suggested revisions aim to improve the methodological clarity, internal coherence, and analytical rigor of the manuscript. Incorporating these adjustments tends to strengthen the study's contribution to the field of public health and social sciences in health.

**Reviewer #2:** The manuscript is of excellent quality, well-written, and presents robust research with findings of great importance. I suggest, if possible, a brief reflection on the role of men in care practices (even if secondary or different) or on the participants' perceptions regarding this could add an extra layer of complexity to the gender analysis. These suggestions only aim to improve a work that already stands out for its rigor and relevance. The publication of this article will certainly contribute to a deeper and more respectful understanding of quilombola health practices and to the advancement of more equitable and culturally sensitive health policies in Brazil.

**Do you want your identity to be public for this peer review?** For information about this choice, including consent withdrawal, please see our Privacy Policy

Reviewer #1: No

Reviewer #2: **Yes:** Pedro Agnel Dias Miranda Neto

---

## [Author Response · Author response to Decision Letter 1]

31 Jan 2026

Dear Editor and Reviewers,

We would like to thank the Editor and the Reviewers for their careful reading of the manuscript and for their valuable and constructive comments. We greatly appreciate the recognition of the relevance, rigor, and social importance of this study. All suggestions were carefully considered, and the manuscript was revised accordingly to improve methodological clarity, conceptual precision, and internal consistency. All modifications are highlighted in red in the revised manuscript.

REVIEWER #1

General Comments

The manuscript was revised to improve methodological transparency and conceptual coherence, particularly in the Materials and Methods and Discussion sections.

Major Comments

1. Redundancy in the description of the hermeneutic-dialectical approach

The two redundant paragraphs were condensed into a single, more concise paragraph to improve clarity and flow.

2. Insufficient description of the statistical dimension of IRAMUTEQ

The Methods section was expanded to explicitly describe the chi-square test used in the Descending Hierarchical Classification, clarify the meaning of the p-value, and explain how statistically significant associations supported qualitative interpretation.

3. Missing Table 1

Table 1 (Sociodemographic characterization) was included in the Results section, and the text was revised for consistency.

4. Exclusion criteria and stigmatizing language

The exclusion criteria were revised to avoid stigmatizing language, specifying cognitive or communicative limitations rather than mental health conditions.

Minor Comments

Duplicate references were corrected, conceptual precision regarding Minayo (2021) was improved, missing references were added, and the use of Habermas and Gadamer was refined.

Final Remarks

We sincerely thank the reviewers for their valuable contributions, which strengthened the manuscript.

Sincerely,

The Authors

---

## [Decision Letter · Decision Letter 1]

5 Feb 2026

Care practices, popular knowledge, and health promotion among quilombola woman in Brazil

PONE-D-25-63692R1

Dear Dr. Gonçalves,

We’re pleased to inform you that your manuscript has been judged scientifically suitable for publication and will be formally accepted for publication once it meets all outstanding technical requirements.

Kind regards,

Rafael Galvão de Almeida, PhD.

Academic Editor

PLOS One

Additional Editor Comments (optional):

Reviewers' comments:

Reviewer's Responses to Questions

**Comments to the Author**

Reviewer #1: All comments have been addressed

2. Is the manuscript technically sound, and do the data support the conclusions?

Reviewer #1: Yes

3. Has the statistical analysis been performed appropriately and rigorously?

Reviewer #1: Yes

4. Have the authors made all data underlying the findings in their manuscript fully available?

Reviewer #1: Yes

5. Is the manuscript presented in an intelligible fashion and written in standard English?

Reviewer #1: Yes

Reviewer #1: (No Response)

**Do you want your identity to be public for this peer review?** For information about this choice, including consent withdrawal, please see our Privacy Policy

Reviewer #1: No

---

## [Editor Report · Acceptance letter]

PONE-D-25-63692R1

PLOS One

Dear Dr. Gonçalves,

I'm pleased to inform you that your manuscript has been deemed suitable for publication in PLOS One. Congratulations! Your manuscript is now being handed over to our production team.

Kind regards,

on behalf of

Dr. Rafael Galvão de Almeida

Academic Editor

PLOS One